# An Osteotomy Tool That Preserves Bone Viability: Evaluation in Preclinical and Clinical Settings

**DOI:** 10.3390/jcm11092536

**Published:** 2022-04-30

**Authors:** Oded Bahat, Xing Yin, Stefan Holst, Ion Zabalegui, Eva Berroeta, Javier Pérez, Peter Wöhrle, Norbert Sörgel, John Brunski, Jill A. Helms

**Affiliations:** 1Private Practice, Beverly Hills, CA 90210, USA; 2Division of Plastic and Reconstructive Surgery, Department of Surgery, Stanford School of Medicine, Stanford, CA 94305, USA; yinxing@scu.edu.cn (X.Y.); brunsj6@stanford.edu (J.B.); jhelms@stanford.edu (J.A.H.); 3Nobel Biocare Services AG, 8058 Zurich, Switzerland; stefan.holst@nobelbiocare.com; 4Private Practice, 48001 Bilbao, Spain; ion@periopixel.com; 5Private Practice, 48008 Bilbao, Spain; eva.berroeta@clinicaevaberroeta.com; 6Private Practice, 27002 Lugo, Spain; javier@oral-design.es; 7Private Practice, Newport Beach, CA 92660, USA; pswohrle@mac.com; 8Private Practice, 80333 Munich, Germany; info@privatpraxis.de

**Keywords:** osteotomy, site preparation, dental implant

## Abstract

The main objectives of this work were to assess the efficiency, ease-of-use, and general performance of a novel osseoshaping tool based on first-user clinical experiences and to compare these observations with preclinical data generated in rodents using a miniaturized version of the instrument. All patients selected for the surgery presented challenging clinical conditions in terms of the quality and/or quantity of the available bone. The presented data were collected during the implant placement of 15 implants in 7 patients, and included implant recipient site (bone quality and quantity) and ridge evaluation, intra-operative handling of the novel instrument, and the evaluation of subsequent implant insertion. The instrument was easy to handle and was applied without any complications during the surgical procedure. Its use obviated the need for multiple drills and enabled adequate insertion torque in all cases. This biologically driven innovation in implant site preparation shows improvements in preserving vital anatomical and cellular structures as well as simplifying the surgical protocol with excellent ease-of-use and handling properties.

## 1. Introduction

Over the past decades, some components of implant systems have undergone radical design alterations [1], yet other features have remained relatively unchanged. One such feature is the utilization of high-speed drilling to prepare implant recipient sites. Conventional site preparation typically involves the use of multiple drills, each with high rotational velocities, to cut bone [2]. Since high-speed drills generate heat, irrigation is required to cool the bone and to remove the osseous coagulum produced by drilling [3,4]. There may be less of a perceived need for improvements in site preparation because dental implants typically enjoy high rates of success [5,6].

Patient expectations, however, have changed the landscape of Implantology. In previous decades, site preparation was followed by the placement of implants that were submerged [7]. Any damage to the bone produced by drilling would, therefore, undergo repair while the implant was free of masticatory loading. Now, patients prefer implant-supported restorations in fewer visits with shorter wait times [8]. To make this possible, peri-implant bone must provide immediate and continuous support for an implant. Clinicians are also increasingly using implant restorations in more challenging conditions, such as their placement into fresh extraction sockets [9] or their placement in sites with limited bone volume [10]. The lower success rates associated with these more challenging conditions prompted a reconsideration of site preparation techniques.

An important consideration in any new design innovation is to understand the limitations of the current system. The goal of site preparation is to rapidly and efficiently remove bone and provide an osteotomy that promotes early cell attachment and matrix deposition [11,12]; however, current protocols leave behind a cut edge that is devoid of viable osteocytes [13,14]. Both thermal and mechanical trauma kill osteocytes [15,16,17,18], but current drill designs necessitate a high rotational velocity to efficiently cut hard material [2]. The preservation of osteocyte viability is of utmost importance [19,20,21] because signaling from osteocytes is key to the recruitment and proliferation of new osteoblasts [14].

Another limiting feature of conventional drilling is the need for irrigation. Although irrigation reduces some of the heat generated by high-speed drilling [3,22,23], it also removes bone chips, connective tissue stroma, and blood. This osseous material is sometimes referred to as osseous coagulum [24], and other times it is referred to as bone debris [14], but regardless of its name, it has proven osteogenic potential [25,26,27,28]. Osseous coagulum is often times collected from drill flutes and stored ex vivo until it can be placed in gaps around an implant [29], but as with any transplanted tissue, ex vivo storage impacts cell viability, and the longer the material is kept outside the body, the more extensive cell death becomes [30].

With these limitations in mind, an optimized osseoshaping tool, named OsseoShaper, was designed to cut bone at low (<50 rpm) speeds. This low rotational velocity generates minimal heat [25,31], and in preclinical animal models, it creates minimal osteocyte death [25] compared to conventional drills. Minimal osteocyte death translates into minimal peri-implant bone resorption; thus, the peri-implant bone structure is maintained [14,25]. The use of irrigation is eliminated when cutting with this new tool; consequently, preclinical data demonstrated that most of the osseous coagulum is retained in the site [14,25]. This osseous coagulum directly contributes to faster peri-implant bone formation [14,26]. These data from animal models supported the development of novel implant site preparation instruments for clinical use.

In the novel implant site preparation protocol, the osteotomy is created in a two-step sequence that starts with the pilot drill and is followed by the application of the osseoshaping tool (Figure 1). The pilot drill creates the initial osteotomy according to the planned depth and angulation of the implant. In the next step, the osseoshaping tool is used to enlarge osteotomy. This novel instrument is threaded, length-specific (i.e., it matches the length of the implant), and is optimized for insertions and removal at a low speed of 50 rpm. Its use replaces the conventional twist drill and twist step drill protocols. In rare cases, such as very dense bone occasionally encountered in the mandible, the instrument cannot be fully seated using the torque of 40 Ncm, and osteotomy requires the use of additional instruments. These include a dense bone optimized version of the osseoshaping tool, and when it cannot be fully seated at 40 rpm, a dense bone drill is used.

The work described in this manuscript assessed the efficiency, ease-of-use, and general performance of the novel osseoshaping tool in a clinical setting, particularly in challenging conditions, by documenting first-user experiences upon market introduction. In parallel, preclinical data were presented that document in rodent models the bone response observed with a miniaturized version of the osseoshaping tool, with the overall goal being to assess the performance of the tools in vivo. The rodent model was selected to closely resemble the bone health of the average dental implant patient with the median age of >50 years [32,33] and tendency toward osteopenia and osteoporosis [34].

## 2. Materials and Methods

### 2.1. Clinical Case Series

This retrospective case series includes patients who had been treated according to the Nobel Biocare N1 system concept, including the novel osseoshaping tool (OsseoShaper 1; Nobel Biocare AB, Göteborg, Sweden), in June 2018 at a private practice in Munich, Germany, and in April 2019 at a private practice in Bilbao, Spain. The case documentation was collected during the introduction phase of this new implant system concept. Standard exclusion criteria for treatment with dental implants were applied, i.e., patients were excluded only in cases in which elective surgery was contraindicated. Clinically challenging implant recipient sites with compromised anatomy were purposely selected (O.B., I.Z., N.S., and P.W.). Specifically, the series included healed sites with compromised bone and immediate post-extractive sites, as well as sites with deficient bone volume, which necessitated the simultaneous surgical reconstruction of bone and soft tissue during implant placement. A signed informed consent for treatment and documentation was obtained from each patient.

The pre-operative analysis consisted of medical clearance, the evaluation of the occlusal, restorative and periodontal statuses, and radiological evaluation including CBCT. The treatment plan also included the details of the provisionalization procedures intended for both pre- and post-surgery, as well as surgical guide designs for the sites located in the esthetic zone.

Data extraction included information on patient demographics (age and gender), surgical site characteristics (FDI position and bone quality and quantity according to Lekholm and Zarb classification [35]), and details of implant placement protocol, including feedback on handling and ease-of-use. This case series report follows CARE (Consensus-based Clinical Case Reporting Guideline Development) guidelines.

#### Surgical and Restorative Procedures

The surgeries were performed under local anesthetic or, when indicated, an intravenous sedation was used. Full thickness flaps were raised at all sites, with the incisions made slightly lingual to the mid ridge in order to retain adequate soft tissue volume, quality, and dimensions. Infected sites as well as recent and fresh extraction sites were debrided mechanically and chemically. Implant site preparation and implant placement followed manufacturer’s recommendations (Figure 1). When augmentation was needed, it was performed in a multi-layer fashion with the internal layer of autogenous bone, mid-layer of allograft or xenograft, and a superficial layer with resorbable membrane alloderm or L-PRP. A simultaneous sinus lift procedure was performed in one patient following the crestal approach. All flaps were advanced and closed primarily without tension, with either chromic gut, vicryl rapid, or PTFE sutures.

None of the implants were loaded immediately. Submerged or open healing (with healing caps) according to a two-stage protocol was applied. Fixed or bonded provisional restorations were used as interim prostheses in the esthetic zone. Implants were loaded 3 to 6 months post-implant placement. After the healing period, implants were uncovered through a small crestal incision, enabling the removal of the implant cover screw. The incision for uncovering was made slightly lingual to the mid ridge in order to retain the dimensions of the soft tissue. Established prosthetic protocols were applied for customized, in-lab manufactured CAD/CAM ceramic restorations that are adhesively bonded to prefabricated standard restorative interfaces (Universal Abutment, N1 TCC, or On1 Universal Abutment, all Nobel Biocare AB).

### 2.2. Animal Model Experiments

#### 2.2.1. Rodent Information

All rodents underwent an ovariectomy (OVX) procedure following the protocol described by Kalu et al. to more closely reflect the general skeletal health of a middle-aged human population [36]. To further align the animal model with patients in this study, where dental implants were placed in healed sites, the animals underwent an extraction of bilateral maxillary first molars in parallel with the OVX procedure. After eight weeks, rats were then randomly assigned to treatment groups outlined in Table 1.

#### 2.2.2. Micro-Computed Tomography (µCT) in Rodents

Scanning and analyses followed published guidelines [37]. In brief, after fixation in 4% PFA at 4 °C overnight, samples were transferred to 70% ethanol solution for µCT scanning prior to decalcification. Scanning and reconstruction were performed using a µCT tomography data-acquisition system (VivaCT 40, Scanco, Brüttisellen, Switzerland) at 10.5 μm voxel size (70 kV, 115 μA, 300 ms integration time), while analysis software (Scanco was used for bone morphometry. Multiplanar reconstruction and volume rendering were performed with Avizo (FEI, Hillsboro, OR, USA) and ImageJ (NIH, Bethesda, MD, USA) software, and the images were then imported into Adobe Photoshop and Illustrator (Adobe, San Jose, CA, USA).

#### 2.2.3. Osteotomy Site Preparation in Rodents

Rodents were randomly assigned to one of two groups: the first group in which the osteotomy was prepared using conventional round drills and a second group where osteotomy preparation was achieved using a miniaturized version of the novel osseoshaping tool. Rodents were anesthetized as described above; then, a full thickness periosteal flap was elevated at a healed maxillary first molar extraction site. A pilot 1.0 mm drill hole generated using a handpiece (KaVo Dental, Uxbridge, UK) and saline irrigation was stepwise enlarged with progressively larger drill diameters (Table 2). In the novel protocol, the pilot drill hole was enlarged using a downscaled prototype of the osseoshaping tool without irrigation, resulting in the same final maximum diameter as the one achieved with the conventional surgical drill protocol. Drill speeds were adjusted to result in the same radial velocity for all drills and to compensate for slightly different diameters. Each osteotomy was created with a new drill. After implant site preparation, tension-free primary closure of the periosteal flap was achieved using tissue glue (VetClose, Henry Schein, Cleveland, OH, USA).

#### 2.2.4. Rodent Tissue Collection and Processing

Tissues were collected at post-osteotomy day (POD) 0.5 to evaluate micro-damage and programmed cell death associated with surgical drilling and at POD3 and POD7 when new bone formation can be visualized. Rodents were euthanized, their entire maxillae were dissected, and the other tissues were removed and fixed with 4% paraformaldehyde (PFA) at 4 °C overnight. Samples were then decalcified in ethylene diamine tetra-acetic acid (EDTA), embedded in paraffin, and sectioned at 8 μm thickness. Following de-paraffinization in Citrisolv (Decon Labs, Inc., King of Prussia, PA, USA) and hydration in a series of decreasing concentrations of ethanol, tissue sections were further analyzed as described below.

#### 2.2.5. Histology

To visualize bone, the sections were stained with aniline blue as follows. Sections were incubated in a saturated solution of picric acid, transferred to a 5% phosphotungstic acid solution, and stained in 1% aniline blue. After dehydration, slides were mounted using Permount (Fisher Scientific, Hampton, NH, USA).

#### 2.2.6. Programmed Cell Death

TUNEL staining (Roche Diagnostics GmbH, Mannheim, Germany) to visualize programmed cell death was performed according to manufacturers’ guidelines. Sections were deparaffinized, rehydrated, and permeabilized for 8 min. After the addition of the TUNEL reaction mixture, sections were then incubated at 37 °C in the dark. For each experimental condition, sections from 4–6 different samples were analyzed. Each section was photographed with a Leica digital image system at 20x magnification, and the number of osteocytes labeled with TUNEL was counted and grouped based on the distance away from the edge of the osteotomy. The corresponding area for each group was then calculated. The number of apoptotic cells per unit area was calculated by dividing the number of apoptotic cells relative to the corresponding area (cell/mm^2^).

#### 2.2.7. Tartrate-Resistant Acid Phosphatase (TRAP) Activity

Bone resorption was visualized with TRAP staining using a leukocyte acid phosphatase staining kit (catalog #386A-1KT, Sigma-Aldrich, St. Louis, MO, USA). TRAP-stained tissue sections were photographed using a Leica digital image system at 10x magnification, and osteoclasts visualized with TRAP staining and within the radial zone of 300 µm from the cut edge were identified to calculate the TRAP^+ve^ ratio, which is expressed as the ration of TRAP^+ve^ pixels to the total pixels in the region of interest.

#### 2.2.8. Immunostaining

Sections underwent immunostaining procedures to localize cells that had initiated differentiation down an osteogenic lineage within osteotomies [38]. After deparaffinization, endogenous peroxidase activity was quenched by 3% hydrogen peroxide for 5 min and then washed in PBS. Sections were then blocked with 5% goat serum (Vector S-1000) for 1 h at room temperature, followed by incubation with the primary antibody overnight at 4 °C, and then washed in PBS. The primary antibodies used in this study were Osterix (1:1200; ab22552, Abcam, Cambridge, UK) and Cathepsin K (1:200; ab19027, Abcam). Samples were incubated with corresponding biotinylated secondary antibodies (Vector BA-x) for 30 min; then, they were washed in PBS, and staining was visualized with an avidin/biotinylated enzyme complex (Kit ABC Peroxidase Standard Vectastain PK-4000) incubated for 30 min and a DAB substrate kit (Kit Vector Peroxidase substrate DAB SK-4100).

#### 2.2.9. Histomorphometric Analyses

Histomorphometric analyses were performed using ImageJ software (NIH, Bethesda, MD, USA). New bone formation in the osteotomy site over time was quantified using a minimum of four osteotomy sites with a minimum of six aniline blue-stained histologic sections that spanned the distance from the furcation to the apex. Each section was photographed using a Leica digital image system at 20x magnification, and they were analyzed for new bone formation by dividing the number of aniline blue^+ve^ pixels within an osteotomy by the number of the total pixels in the same osteotomy area.

#### 2.2.10. Statistical Analyses

Results are presented in the form of mean ± standard deviation. *t*-tests and paired *t*-tests were performed with significance level set at *p* < 0.05. All statistical analyses were performed with SPSS software.

## 3. Results

### 3.1. Clinical Cases

This clinical case series included 7 patients who received 15 implants. The details of each case are listed in Table 3.

Figure 2 shows the pre-treatment CBCT evaluation of two patients, one of which required treatment in the esthetic zone with simultaneous bone and soft tissue augmentation (Figure 2a) and one who had restoration planned with two implants using the simultaneous sinus lift procedure (Figure 2b). Examples of implant recipient sites after flap elevation are shown in Figure 3. Most (13 of 15) recipient sites had soft (class 3) or very soft (class 4) bone. During implant site preparation procedures, the osseoshaping tool was fully seated in all but three cases: two of which had very soft bone and where the instrument was advanced only halfway, and one case with harder bone that required the use of a dense bone-optimized osseoshaping tool. Figure 4 shows images of the novel instrument upon withdrawal from the osteotomy. The use of the osseoshaping tool was followed immediately by implant insertion. Of the 15 sites, 7 were augmented due to deficient vertical and/or horizontal ridge dimension. All implants were loaded in a delayed protocol. At the four sites in the esthetic zone, fixed or bonded restorations were used. Thirteen sites underwent submerged healing according to a two-stage protocol while the remaining two sites were restored in a two-stage protocol with an open healing approach, using healing caps during the osseointegration period. The surgical procedure and the images collected at the follow-up visit for two selected cases are shown in Figure 5, Figure 6 and Figure 7.

In clinical applications, the novel osseoshaping tool instrument shows good handling properties. The instrument followed the initial osteotomy created by the pilot drill, without any wobbling at the 50 rpm speed recommended by the manufacturer. As expected, the low noise and vibration had a positive impact on patient comfort. The osseoshaping tool was stable upon insertion, including insertions in irregular bone crests. Most bone chips created during osteotomy formation appeared to remain within the site, which is likely due to absence of irrigation. In most cases, the entire procedure consisted of only three steps: pilot drill; the osseoshaping tool; implant insertion.

Implants inserted into the sites prepared with the osseoshaping tool followed the osteotomies accurately. For 13 of 15 implants, the final insertion torque was above 35 Ncm, while the remaining two implants achieved a torque of 25 and 31 Ncm.

### 3.2. Evaluation of Rodent Data

To align a preclinical rodent model with a representative clinical case presented here (Figure 2a,b) with regard to bone density, a maxillary first molar was extracted and site preparation was undertaken after.

In clinical cases, osteotomies were produced with a novel osseoshaping tool, whereas in preclinical cases, osteotomies were either produced with a control (conventional high-speed) drill (Figure 8f) or with a miniaturized version of the instrument (Figure 8g). The appearance of conventionally prepared osteotomies differed from those prepared by the novel instrument. The cutting flutes of a conventional drill removes osseous coagulum and bone chips, leaving behind a smooth, glassy surface (Figure 8e). In contrast, osteotomies produced by the novel instrument had a textured appearance, which is attributable to the retention of osseous coagulum in the site (not shown). Site preparations in a rodent model yielded identical results: osteotomies produced with conventional high speed drills had a smooth, glassy surface compared to those generated by the miniaturized instrument, which had a textured surface (Figure 8f,g).

The potential significance of the textured osteotomy surface was explored using the preclinical rodent model. Histology was performed and compared to conventional osteotomies that were devoid of bone chips, and osteotomies created with the osseoshaping tool were occupied by osseous coagulum (Figure 8j). The presence of osseous coagulum was accompanied by an increase in the expression of osteogenic proteins, including Runx2 and Osterix. Additionally, these osteotomies had significantly fewer apoptotic osteocytes compared to conventional osteotomies. 

The zone of death produced during conventional osteotomy site preparation was accompanied by exuberant bone resorption, and it was visualized using TRAP staining. In comparison, the zone of death produced during osteotomy site preparation using osseoshaping tool was much smaller and was accompanied by significantly less TRAP staining. Taken together, these data demonstrated that osteotomies produced with the miniatured instrument were more viable and more osteogenic than osteotomies produced with conventional cutting tools.

## 4. Discussion

Preservation of bone viability within the implant recipient site is key to promote early and successful osseointegration of a dental implant. Biological and molecular findings have supported the design and development of new site preparation tools and protocols; moreover, in the animal model experiments as well as the clinical cases described here, we show that these innovations facilitate a robust and healthy tissue response within osteotomies.

The resulting novel concept introduces a streamlined, biologically driven site preparation protocol to preserve vital bone and thereby to promote rapid osseointegration. The novel osseoshaping tool has performed well in clinical applications—especially considering the poor bone quality of the sites and challenging ridge anatomy included in this case series. The initial angulation and depth of the osteotomies created with the pilot drill were followed with no change in direction by both the osseoshaping tool and the implant, reliably leading to high insertion torques. 

The healing and integration patterns in the human subjects and the aligned rodent model are comparable and, thus, provide additional information on the molecular and cellular response to implant site preparation with this novel protocol. The osteogenic potential is increased in osteotomies created with this protocol: Osteocyte cell death is reduced, and the osseous coagulum produced by low drilling speed is retained within the osteotomies, while the congruence between the osseoshaping instrument and the implant leads to the stabilization of the coagulum around the implant. 

Compared to a conventional high-speed drill protocol, the use of the osseoshaping tool reduces guesswork regarding bone quality at the implant recipient site. Torques generated using the novel instrument guide the surgical workflow, thus providing assistance with the evaluation of bone quality, and can help anticipate implant stability [39]. This feature is possible because the microgeometry and active areas of each osseoshaping tool are individualized to respective implant lengths, which is why they are delivered for single use applications and are co-packed with the corresponding implant.

The other benefits of the novel protocol confirmed by this work are the increased patient comfort due to the reduction in noise and vibration [40] and the overall reduction in the number of surgical entries. This is in agreement with the results reported in a recent retrospective study on the use of the osseoshaping tool in which the authors found that, in 94.5% of all implants placed, only one instrument was used in addition to the initial pilot drill prior to implant insertion [41]. Similarly, in this case series, 14 of 15 implant insertions were performed after a two-step osteotomy formation.

The main limitations of this study are associated with the low number of cases and the lack of evaluation of the osseointegration patterns associated with the novel concept system. The low number of cases reflects that case documentation was collected during the introduction phase of this new implant system concept and that the cases were selected for compromised bone conditions. Further research, particularly the assessment of the clinical and radiographic performance, is needed to confirm these initial findings.

Since its inception in the 1960s, implant dentistry innovations have largely focused on implant macrogeometry while little attention has been given to implant site preparation protocols and tooling. Recently, biological basic research has provided key insights into tissue damage and following healing patterns in response to conventional site preparation procedures. Drawing from these findings and the associated opportunities to improve current protocols and instrumentation, the newly developed osseoshaping tool concept eliminates many shortcomings and improves the biological response to the damage associated with implant site preparation. However, it is important to note that the improvements introduced by this novel site preparation protocol are designed to work in concert with the new biologically driven implant design. In this innovative concept system, an osteotomy is created as an implant recipient site that mimics the implant macroshape for each and every implant size. Further clinical studies are needed to confirm the biological benefits, particularly in terms of peri-implant bone response, of this innovation.

## 5. Conclusions

The novel osseoshaping tool introduces a biologically driven innovation in implant site preparation that shows improvements in preserving vital anatomical and cellular structures as well as simplifies the surgical protocol. The instrument was easy to handle, and its use was not associated with any complications during the surgical procedure while eliminating the need for multiple drills. Adequate implant insertion torque was achieved in all cases. Further evaluation in clinical studies is required to assess the osseointegration of implants following implant site preparation wit this novel protocol.

## Figures and Tables

**Figure 1 jcm-11-02536-f001:**
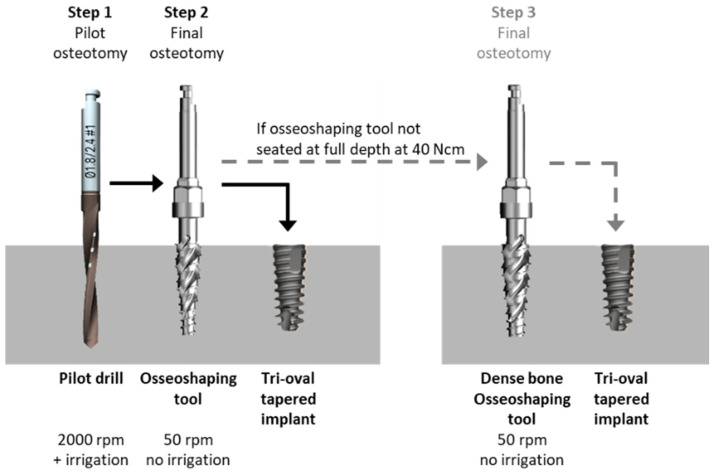
Novel osteotomy formation protocol with the osseoshaping tool.

**Figure 2 jcm-11-02536-f002:**
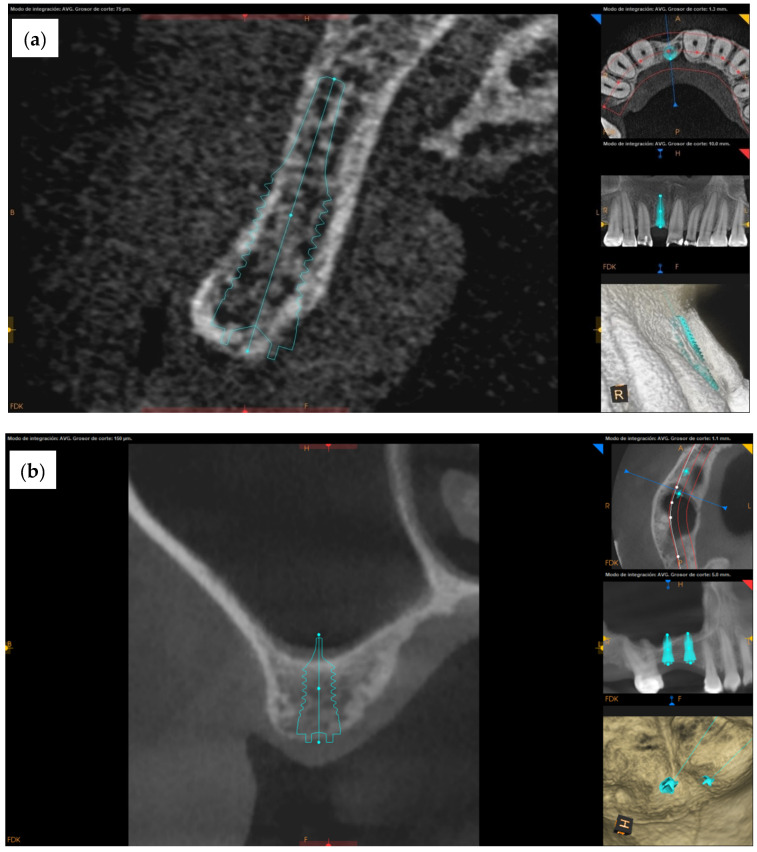
Pre-operative diagnostics and implant treatment plan. CBCT images prior to surgery from two selected clinical cases. (**a**) A 38-year-old female with a missing maxillary central incisor (FDI position 11) and a horizontally compromised ridge, planned for a single implant-supported restoration with simultaneous guided bone regeneration. (**b**) A 47-year-old female with two missing teeth at FDI positions 15 and 16, planned for placements of two implant with simultaneous sinus floor elevation (crestal approach).

**Figure 3 jcm-11-02536-f003:**
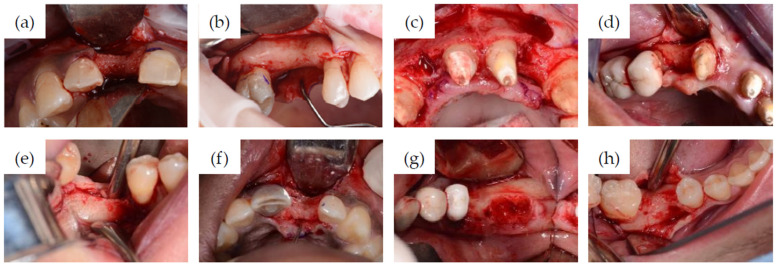
Selected clinical views of implant recipient sites after full-thickness flap elevation (**a**–**h**). (**a**,**b**), flap elevation in patients shown in Figure 2a,b, respectively.

**Figure 4 jcm-11-02536-f004:**
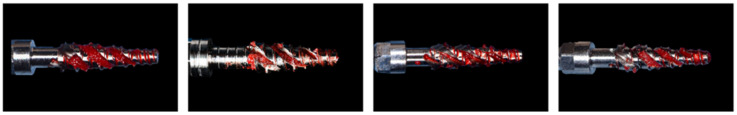
The novel osseoshaping tool upon withdrawal from the osteotomy site. Note the autogenous bone within the threads of the instrument.

**Figure 5 jcm-11-02536-f005:**
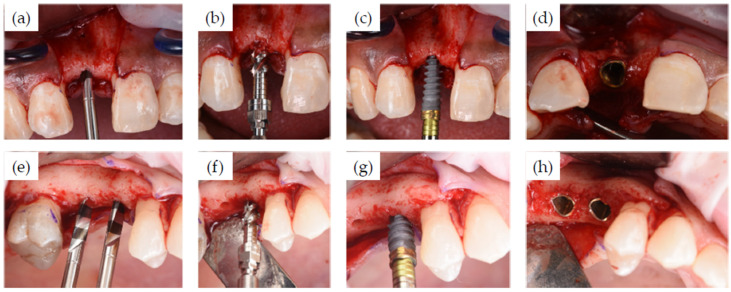
Implant site preparation and implant insertion. (**a**–**d**) Implant site preparation with the pilot drill (**a**), followed by the osseoshaping tool (**b**), and implant insertion (**c**) with the final insertion torque of 55 Ncm. (**d**) Occlusal view of implant in place. (**e**–**h**) After establishing the initial osteotomy with the pilot drill (**e**), the site was prepared with the osseoshaping tool (**f**). The sinus lift procedure was performed using osteotomes, and the implant was inserted (**g**) with a final insertion torque of 70 Ncm. (**h**) Occlusal view immediately after post-implant placement.

**Figure 6 jcm-11-02536-f006:**
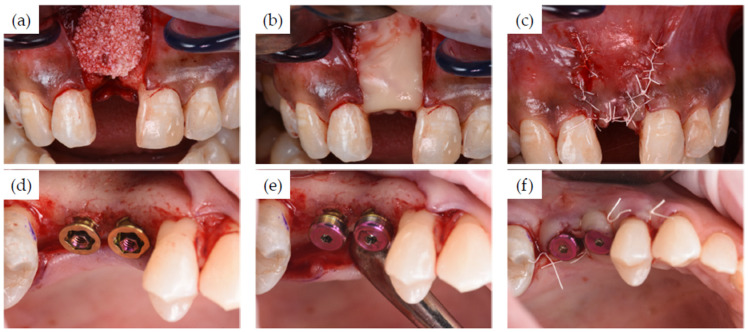
Flap closure. (**a**–**c**) An allograft and autogenous bone graft was placed (**a**) and covered with L-PRF membrane (**b**), and the flap was sutured (**c**). (**d**–**f**) A transmucosal base was placed on each implant at time of surgery (**d**) and connected to a healing cap (**e**). (**f**) Flap closure to facilitate the shaping of the mucosa by healing caps and the base.

**Figure 7 jcm-11-02536-f007:**
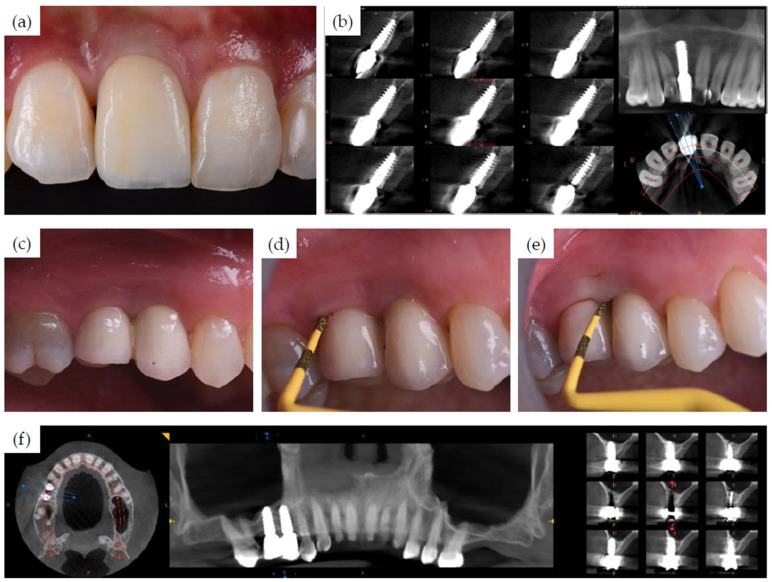
Clinical and radiographic follow-up. Clinical view (**a**) and CBCT (**b**) of the single implant-supported restoration of the missing maxillary central incisor at 32 months post implant insertion. (**c**–**f**) Two single implant-supported restorations in the maxilla at 31 months post implant insertion. Note the healthy peri-implant mucosa upon passing the dental probe along the mucosal margin (**d**,**e**).

**Figure 8 jcm-11-02536-f008:**
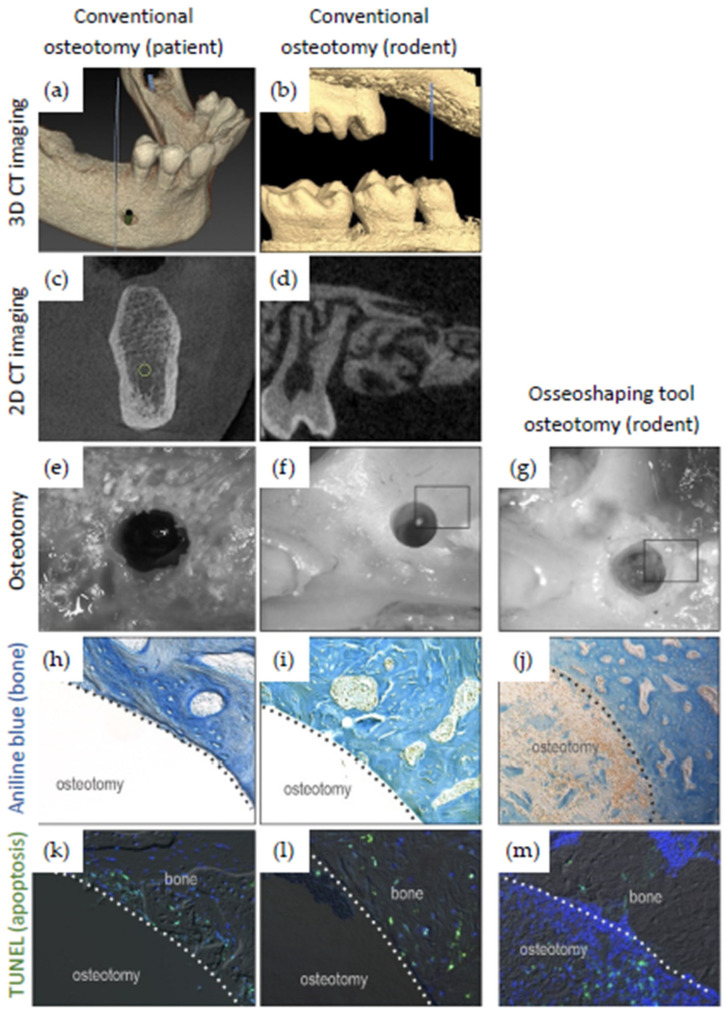
Comparison of osteotomies created following a conventional site preparation protocol (**left** and **middle columns**) and using the osseoshaping tool (**right column**). (**a**–**d**) Pre-operative patient CBCT (**a**,**c**) and rodent µCT (**b**,**d**) demonstrating the similar bone quality of the implant recipient site in human versus rodent. (**e**–**g**) Osteotomies created with conventional drills (**e**,**f**) and the osseoshaping tool (**g**). Note the irregular walls around the latter osteotomy (**g**). (**h**–**m**) Osteotomy sections stained for aniline blue to reveal bone and with TUNEL to reveal cell death (apoptosis) from sites prepared conventionally or with the osseoshaping tool. Application of the osseoshaping tool results in the visible presence of the osseous coagulum (**j**) and decreased levels of apoptosis (**m**). Dotted lines show the edge of the osteotomy.

**Table 1 jcm-11-02536-t001:** Preclinical experimental groups.

Experimental Group	Sample Size	Tool	Rotational Velocity (rpm)	Irrigation	Time Points for Data Analysis	Analyses Performed
Conventional osteotomy site preparation	18	Final OsteoMed drill	1350	yes	0.5 days 7 days	µCT, histology, IHC
N1 site preparation	18	Osseo-Shaper	50	no	0.5 days 7 days	µCT, histology, IHC

µCT, micro-computed tomography; IHC, immunohistochemistry.

**Table 2 jcm-11-02536-t002:** Drill parameters.

Company	Osteotomy Instrument	External Diameter (mm)	Revolutions per Minute (rpm)	Irrigation
OsteoMed	Pilot drill	0.8	2000	yes
	2nd drill	1.0	1600	yes
	Final drill	1.2	1350	yes
Nobel Biocare	Pilot drill	0.8	2000	yes
	Miniaturized version of osseoshaping tool	crestal = 1.33 apical = 0.525	50	no

**Table 3 jcm-11-02536-t003:** Summary of patient and surgical characteristics included in the clinical case series.

Patient Number	Patient Characteristics		Implant Recipient Site Characteristics				Surgical Details		
	Age	Gender	FDI Position	Bone Quality and Quantity	Additional Notes on Bone Conditions	Site Type	Insertion Torque (Ncm) of the Osseoshaping Tool	Implant Insertion Torque (Ncm)	Simultaneous Bone Augmentation
1	38	female	11	3, B	buccal concavity	healed	Not reported	45	yes
2	47	female	15	4, C	Sinus lift	healed	Not reported	45	no
			16	4, C	Sinus lift	healed	Not reported	45	no
3	58	female	12	4, B	buccal concavity	recent extraction (12 weeks)	40 *	61	yes
			14	4, B	-	healed	10	25	no
			22	4, B	large concavity	healed	40 *	39	yes
			37	4, B	-	healed	19	55	no
			38	4, B	-	extraction	5	50	yes
4	64	female	46	2, C	narrow ridge	healed	34/21 **	60	no
5	60	female	46	2, C	-	healed	23	50	yes
			47	3, D	-	healed	21	50	yes
			36	3, B	-	healed	16	37	no
			37	3, B	-	healed	16	31	no
6	18	male	21	3, B	large concavity, 6 mm lingual defect	healed	35	55	yes
7	26	male	36	3, B	buccal concavity	healed	38	55	no

* The instrument was inserted only halfway due to very soft bone conditions. ** Dense bone instrument was used.

## Data Availability

The data presented in this study are available upon request from the corresponding author.

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
