# Peer review of "An Osteotomy Tool That Preserves Bone Viability: Evaluation in Preclinical and Clinical Settings"

_jcm, 2022, doi:10.3390/jcm11092536_

Round 1
Reviewer 1 Report
Excellent study on an interesting topic, well-written manuscript. Only suggestion: limit the "conclusions" to the real "conclusions" related to the outcomes of the study and eliminate unrelated information (this could be moved to the end of the "discussion". Make sure conclusions in the abstract are the same as in the main text.
Reviewer 2 Report
The authors have shown that the newly developed osseoshaping tool concept eliminates many of the shortcomings and improves the biological response to the damage associated with implant site preparation by using both clinical and rodent models. Definitely, the biggest drawback of this study is a very small number of cases, and most of the cases are females (5:2).
What is the implications of this gender biasness in terms of selecting patients? In a rodent model, the authors performed ovariectomy. However, it is not very clear why? Is it because most of the human cases were females and of the menopausal age? If that is true, then one should realize that in menopausal women all the phenotypes are not merely due to the absence of ovarian hormones but may be age-related effects, for which the authors have not provided any solid explanation.
Overall, the paper is good but lacks the rationale for the selection of experimental protocol.
Reviewer 3 Report
GOOD MORNING. FIRST OF ALL I WANT TO THANK THE PUBLISHER FOR ALLOWING ME TO COLLABORATE IN THE ANALYSIS AND REVISION OF THIS ARTICLE.
SECONDLY, I WANT TO CONGRATULATE THE AUTHORS OF THIS ARTICLE BECAUSE I CONSIDER THAT THE SUBJECT WE DEAL WITH IS OF HIGH IMPORTANCE AND RELEVANCE IN THE AREA OF IMPLANTOLOGY.
AS A RECOMMENDATION, I CONSIDER THAT THE SAMPLE OF IMPLANTS PLACED THROUGH THE USE OF THIS NEW TECHNIQUE SHOULD BE INCREASED.
